# Circulatory Inflammatory Mediators in the Prediction of Anti-Tuberculous Drug-Induced Liver Injury Using RUCAM for Causality Assessment

**DOI:** 10.3390/biomedicines9080891

**Published:** 2021-07-25

**Authors:** Cheng-Maw Ho, Chi-Ling Chen, Chia-Hao Chang, Meng-Rui Lee, Jann-Yuan Wang, Rey-Heng Hu, Po-Huang Lee

**Affiliations:** 1Department of Surgery, National Taiwan University Hospital and College of Medicine, Taipei 10617, Taiwan; miningho@ntu.edu.tw (C.-M.H.); rhhu@ntu.edu.tw (R.-H.H.); pohuang1115@ntu.edu.tw (P.-H.L.); 2Graduate Institute of Clinical Medicine, National Taiwan University, Taipei 10617, Taiwan; chlnchen@ntu.edu.tw; 3Department of Internal Medicine, National Taiwan University Hospital, Hsinchu Branch, Hsinchu City 300, Taiwan; ntudcm@gmail.com (C.-H.C.); sheepman1024@gmail.com (M.-R.L.); 4Department of Internal Medicine, National Taiwan University Hospital and College of Medicine, Taipei 10617, Taiwan

**Keywords:** drug-induced liver injury, hepatotoxicity, inflammatory mediators, prediction model, tuberculosis, RUCAM (Roussel Uclaf Causality Assessment Method)

## Abstract

Background: Anti-tuberculous (TB) medications are common causes of drug-induced liver injury (DILI). Limited data are available on systemic inflammatory mediators as biomarkers for predicting DILI before treatment. We aimed to select predictive markers among potential candidates and to formulate a predictive model of DILI for TB patients. Methods: Adult active TB patients from a prospective cohort were enrolled, and all participants received standard anti-tuberculous treatment. Development of DILI, defined as ≥5× ULN for alanine transaminase or ≥2.6× ULN of total bilirubin with causality assessment (RUCAM, Roussel Uclaf causality assessment method), was regularly monitored. Pre-treatment plasma was assayed for 15 candidates, and a set of risk prediction scores was established using Cox regression and receiver-operating characteristic analyses. Results: A total of 19 (7.9%) in 240 patients developed DILI (including six carriers of hepatitis B virus) following anti-TB treatment. Interleukin (IL)-22 binding protein (BP), interferon gamma-induced protein 1 (IP-10), soluble CD163 (sCD163), IL-6, and CD206 were significant univariable factors associated with DILI development, and the former three were backward selected as multivariable factors, with adjusted hazards of 0.20 (0.07–0.58), 3.71 (1.35–10.21), and 3.28 (1.07–10.06), respectively. A score set composed of IL-22BP, IP-10, and sCD163 had an improved area under the curve of 0.744 (*p* < 0.001). Conclusions: Pre-treatment IL-22BP was a protective biomarker against DILI development under anti-TB treatment, and a score set by additional risk factors of IP-10 and sCD163 employed an adequate DILI prediction.

## 1. Introduction

Drug-induced liver injury (DILI) is a challenging medical issue because of the obscure nature of liver injury during development and the lack of available biomarkers for early prediction [1]. Ordinarily, liver injury is more often caused by a combination of multiple factors, such as viral hepatitis, alcoholic, or non-alcoholic steatohepatitis, in addition to drugs or complementary medications [2,3,4]. Prevention is the best policy, although difficult.

Significant concern about developing DILI during anti- tuberculous (TB) treatment has been raised because it can result in the modification or discontinuation of anti-TB treatment in about one-sixth of TB patients [5,6] and is difficult to manage, rendering further dissemination of *Mycobacterium tuberculosis*. DILI under anti-TB treatment (anti-TB DILI) is the most common cause of DILI and drug-induced acute liver failure in many Asian countries [7] and the second most common cause of drug-induced acute on chronic liver failure in Asia [7]. The incidence of anti-TB DILI varies depending on the definition used to define DILI and varies from 3.4% to 7.3% [7,8].

Studies of biomarkers in DILI can be applied in diagnosis, severity, or prognosis [9] Currently few biomarkers for DILI (including anti-TB medications) are useful for early detection, monitoring, or for diagnosis purposes [9]. Steuerwald et al. had previously compared serum immune analytes (14 cytokines, 7 chemokines, and 6 growth factors) at DILI onset and after 6 months, with healthy controls, among the USA DILI cohort and demonstrated that high levels of expression of cytokines associated with innate immunity are associated with a poor prognosis, whereas high levels of expression of adaptive cytokines are associated with good long-term prognosis and eventual recovery [10] However, their cohort is heterogeneous and not specific to TB medications. In addition, they used the serum of patients who had already developed DILI.

Risk factor analysis of circulating inflammatory mediators, particularly for TB patients, is not adequately addressed, if ever, that would greatly bridge the gap between genetic and epidemiological risk. We hypothesise that pre-treatment plasma biomarkers are associated with the development of DILI in patients with TB. We aimed to analyse selective cytokines, chemokines, growth factors, and regulatory proteins, and relevant collected clinical data to formulate a prediction model of DILI once TB treatment started.

## 2. Patients and Methods

### 2.1. Patients

Adult patients (aged between 20 and 90 years) who were diagnosed with culture-confirmed all-susceptible pulmonary TB, joined previous prospective clinical studies conducted at the National Taiwan University Hospital (NTUH) (Institutional Review Board [IRB] No.: 201303063RINC, 201403048RIND, 201512169RINA) and agreed to share residual pre-treatment blood samples for other later studies. This study was approved by the NTUH IRB (202005100RINC). Among the participants in this study cohort, published data were derived from only fifty-three patients enrolled in IRB No. 201512169RINA in a non-relevant subject [11].

### 2.2. Protocol

Before anti-tuberculous treatment began, aspartate transaminase (AST), alanine transaminase (ALT), total bilirubin (T-Bil), and serological tests for hepatitis B virus (HBV), hepatitis C virus (HCV), and human immunodeficiency virus (HIV) were performed. A liver function test (LFT) was then performed at 2, 4, 6, 8, 12, and 16 weeks after the start of anti-tuberculous treatment or whenever symptoms of hepatitis developed, and clinically relevant hepatitis was suspected by the primary care physician during the course of treatment [12] Additional testing would be performed if primary care physicians suggested based on clinical condition.

All participants received standard anti-tuberculosis treatment of daily isoniazid (INH), rifampin (RMP), ethambutol (EMB), and pyrazinamide (PZA) in the first two months and daily INH and RIF for the subsequent four months. The regimen was modified, if necessary, by the primary care physician [12].

### 2.3. Clinical Parameters

Demographic and clinical data, including underlying comorbidities (presence of hepatitis A virus (HAV), hepatitis B virus (HBV), hepatitis C virus (HCV), hepatitis E virus (HEV), human immunodeficiency virus (HIV), diabetes mellitus (DM), hypertension, positive cancer history, autoimmune disease, and receiving haemodialysis), chronic smoking and alcohol use, date of blood sampling and starting anti-TB treatment, and blood test results were collected.

### 2.4. Outcome Definitions of DILI

Baseline abnormal liver function test results were defined as elevated ALT ≥ 2 times the upper limit of normal (ULN) before the commencement of anti-TB therapy. When abnormal liver function panels were noted during follow-up, patients were screened for viral hepatitis (HAV, HEV, and reactivation of HBV), autoimmune hepatitis and enquiry on the use of other hepatotoxic drugs or complementary and alternative medicine [7]. DILI is defined as ≥5× ULN for ALT or ≥2.6× ULN of total bilirubin, based on the Division of AIDS (DAIDS) Table for Grading the Severity of Adult and Paediatric Adverse Events [13] and causality assessment by using RUCAM (Roussel Uclaf Causality Assessment Method) [14].

### 2.5. Assays of Plasma Biomarkers

Inflammatory markers in plasma potentially relevant to TB-associated DILI were assayed using the Invitrogen™ ProcartaPlex™ Human 65-plex panel (Thermo Fisher Scientific, Waltham, MA, USA) kit and enzyme-linked immunosorbent assay (ELISA), including cytokines (IL-22, IL-22 binding protein (IL-22BP), IL-6, IL-10, IL-12p70, IL-17A, IL-23, IP-10 (CXCL10, interferon gamma (IFN-gamma)-induced protein 10)], chemokines [RANTES (CCL5, regulated on activation, normal T cell expressed and secreted), MIG (CXCL9, monokine induced by gamma interferon), MIP (macrophage inflammatory proteins)-1β], macrophage activation markers (CD206 and sCD163), and growth factors (fibroblast growth factor [FGF]-2, platelet-derived growth factor [PDGF]-bb), according to the manufacturers’ recommendations (ELISA kits for IL-22BP, CD206, and sCD163: MyBioSource, San Diego, CA, USA; for other markers: Thermo Fisher Scientific, Waltham, MA, USA).

### 2.6. Statistical Analysis

Data are expressed as mean ± standard deviation, median (interquartile range [IQR]), or number (percentage) when appropriate. Student’s *t*-test, the χ^2^ test, Fisher’s exact test, or the Mann-Whitney U test were used, where appropriate, for the comparison of variables. Cox’s proportional hazard model was used for univariable and stepwise multivariable analyses; the latter was employed to identify prognostic factors associated with the development of DILI during anti-TB treatment and to adjust for potential confounding factors.

Cytokine levels were transformed into binary variables according to Youden’s index [15] before entering the Cox regression analysis. Optimal cut-off values for biomarkers were calculated from a receiver operating characteristic (ROC) curve. Tertiles (three 3-levels) of biomarkers, with cut-off values determined from the non-DILI group, were used in the sensitivity analysis.

Statistical significance was set at *p* < 0.05. Analyses were performed using the Statistical Package for Social Sciences (SPSS)^®^ version 21.0 (SPSS Inc., Chicago, IL, USA).

## 3. Results

### 3.1. Demographics, DILI Correlation, and DILI-Free Survival

The patient flow is shown in Figure 1. From December 2011 to July 2017, 1442 patients were diagnosed with culture-confirmed pulmonary TB. Among the 363 patients who agreed to participate, a total of 240 patients were enrolled in this study after excluding patients who were transferred out later, had drug-resistant TB, and refused to share residual blood samples.

Patient demographics are listed in Table 1. None of the patients had hepatic tuberculosis, HIV, or chronic alcohol use. Nineteen patients (7.9%) met the criteria of DILI (DILI group) (17 hepatocellular type, 1 cholestatic type, and 1 mixed type) during anti-TB treatment, with 3 being judged as highly probable and 16 as probable according to the RUCAM score and hepatologist consultation (Table 2). Most patients (18/19, 94.7%) developed DILI within 70 days of initiating anti-TB treatment. DILI-free survival for 1, 2, 3, 6, and 9 months were 98.7%, 94.5%, 92.7%, 92.0%, and 90.4%, respectively. Compared to those without DILI (non-DILI group), the DILI group were older, had more HBV carriers, had fewer mean dosing days for INH and PZA, and higher initial total bilirubin (all *p* < 0.05). Clinical factors with trends of frequency difference between the groups included more active smoking, fewer dosing days of EMB, and more abnormal initial liver function tests in the DILI group (all *p* < 0.10).

In univariable Cox analysis, age, with a hazard ratio (HR) of 1.03, (95% confidence interval [95% CI], 1.00–1.06), HBV carriers, with an HR of 3.29, (1.25–8.78), and baseline T-Bil, with an HR of 3.46, (2.27–5.30) were clinical risk factors associated with developing DILI after starting TB treatment.

### 3.2. Plasma Biomarkers and DILI Correlation

The plasma levels of the 15 candidate biomarkers are shown in Figure 2. The DILI group had significantly higher levels of IL-6 and IP-10 than the non-DILI group. The optimal cut-off values for these biomarkers, with individual sensitivity and specificity, are displayed in Appendix A.

In univariable Cox analysis, IL-6 (HR, 2.49 (1.00–6.20)), IP-10 (3.21 (1.22–8.44)), CD206 (4.02 (1.17–13.8)), and sCD163 (3.47 (1.15–10.5)) were significant risk factors associated with DILI development (Table 3). IL-22BP (0.32 [0.11–0.88]) was a significant protective factor against the development of DILI. In multivariable Cox analysis, IL-22BP, IP-10, and sCD163 were independent factors associated with DILI development, with respective adjusted HRs of 0.20 (0.07–0.58), 3.71 (1.35–10.21), and 3.28 (1.07–10.06) (Table 3).

### 3.3. IP-10, IL22BP, and sCD163: Patient Characteristics, DILI-Free Survivals, and Sensitivity Analysis

The point of ROC curve maximizing the Youden Index (sum of sensitivity and specificity) was determined as the optimal cut-off value [16]. The cut-off values were 3.9 pg/mL, 520 ng/L, and 1.32 ng/mL for IP-10, IL22BP, and sCD163, respectively. Patient characteristics, stratified by the respective dichotomous levels of IP-10, IL-22BP, and sCD163, are shown in Table 4. Patients with high IP-10 levels (*n* = 96) were older, had underlying hypertension, and the hepatitis C virus, and had higher levels of baseline LFT (AST, ALT, and T-Bil) than those with lower IP-10 levels (all *p* < 0.05) (Table 4). Compared to low levels, for patients with high IL-22BP levels (*n* = 27), more underlying hypertension and less HBV infection were observed; for higher sCD163 levels (*n* = 19), older patients were observed (all *p* < 0.05). Patients with high IP-10, low IL-22BP, and high sCD163 levels had significantly inferior individual DILI-free survival than those in the other groups (Figure 3, upper panel). These trends could be seen in tertiles (Figure 3, middle panel) and the HBV subgroup (Figure 3, lower panel).

### 3.4. Scoring Predicting DILI

An arbitrary scoring system was developed by the summation of dichotomous levels of IP-10, IL-22BP, and sCD163. Namely, the score = IP-10 (0 or 1) + IL-22BP (0 (high) or 1 (low)) + sCD163 (0 or 1). DILI-free survival stratified by three categories of scores (0, 1, and ≥2) is shown in Figure 4A, with higher scores indicating a greater risk of DILI development. The HR for score 1 (score 0 as reference) was 5.15 (1.11–23.80), and for score ≥2, 11.26 (1.90–66.72) after adjusting age and HBV (model 1, Figure 4B). When further adjusting baseline total bilirubin level (model 2), the similar trend was demonstrated with the HR 4.23 (0.85–21.0) for score 1 and 14.45 (2.17–96.12) for score 2. The AUC based on the scoring system was 0.744 (*p* < 0.001), higher than that of IL-22BP, IP-10, and sCD163 and 12 other inflammatory markers listed in Appendix A.

## 4. Discussion

Our study revealed four main findings. First, nearly 8% of patients with culture-confirmed drug-susceptible pulmonary TB developed DILI after treatment, and over 90 % of them occurred within 70 days. Second, advanced age and HBV carriers were two clinical risk factors for DILI development in our prospective cohort. Third, among the 15 plasma biomarker candidates, pre-treatment IP-10, sCD163, and IL-22BP were narrowed down as risk (IP-10 and sCD163) and protective (IL-22BP) predictors of future DILI. Lastly, after adjusting for age and HBV status, a score composed of the three aforementioned biomarkers had a dose-responsive hazard for DILI prediction and employed an AUC of 0.744.

We validated the 19 cases of DILI (3 as highly probable and 16 as probable) cautiously by using RUCAM and hepatologist consultation. Many DILI cases would not be DILI but something else if the cases were not evaluated for causality assessment method such as RUCAM [14] Rathi et al. had used RUCAM prospectively in their paper and identified 40 patients with anti-TB DILI, among all 82 patients with DILI after excluding eight patients for whom DILI was deemed unlikely to be responsible for liver injury [17] RUCAM was appreciated in more than 80,000 DILI cases published worldwide [18]. Education and training on RUCAM should be encouraged to improve the results of the studies and the day-to-day work in pharmacovigilance departments in companies or in regulatory agencies [19]. It is also expected to improve RUCAM with biomarkers or other criteria provided that the validation process replaces expert opinion by robust standards such as those used for the original method [19]. However, RUCAM was not designed for suspected chronic DILI, which is mostly an unrecognized preexisting liver disease [20]. Besides, RUCAM was also not designed when a suspected injury occurs on preexisting liver diseases (such as 6 in our 19 cases), a complex condition where expert hepatologists are required [20].

It is possible to reduce treatment-related complications and morbidity by conducting frequent check-ups for liver function in those with a high risk of DILI and prevent further dissemination of TB transmission due to treatment interruption. A net score combining IP-10, IL-22BP, and sCD163 (although the cut-off points were selected statistically without definite priori biological reasoning), developed in our study, appears to help predict the inflammation perturbation caused by TB medications which predispose TB patients to DILI development. Circulating soluble mediators in TB patients before treatment can potentially reflect the immune-inflammatory state in a particular host. We identified a novel protective (IL-22BP) and two harmful (IP-10 and sCD163) roles associated with DILI development. IP-10, secreted by monocytes, endothelial cells, and fibroblasts in response to IFN-γ [21], is up-regulated in numerous chronic diseases, including hepatitis B/C [22,23] tuberculosis [24], diabetes [25], and autoimmune disorders [26]. Serum IP-10 level had been reported to be associated with the severity of DILI [27]. probably through the downstream T cell-mediated hepatitis [28]. sCD163 may be released from tissue macrophages and monocytes by a metalloprotease-dependent pathway associated with the inflammation-inducible enzyme TNF-α converting enzyme (TACE/ADAM17) [29]. The exact physiological role as an inflammatory mediator is not clear, although some mechanistic studies have suggested inhibition of activated T lymphocyte proliferation [30] and enhancement of pathogen recognition and phagocytosis [31]. Increased plasma levels of sCD163 have been linked to states of low-grade inflammation such as diabetes, obesity, liver disease, tuberculosis, and atherosclerosis [32,33,34,35,36]. In chronic HBV and HCV infection, sCD163 levels increase with incrementing stages of liver inflammation and fibrosis [37] and the highest levels of sCD163 had been described in patients with acute liver failure especially among patients with fatal outcome [38]. Thus, a correlation between sCD163 level and liver disease severity is evident [39]. In summary, both IP-10 and sCD163 are markers associated with TB and liver inflammation and the inclusion of adaptive (IP-10) and innate (sCD163) immunity in the score set is biologically plausible.

The protective role of IL-22BP in DILI is complex. IL-22 is primarily produced at barrier surfaces by T cells and innate lymphoid cells and is crucial to maintain epithelial integrity [40], potentially through mediating the crosstalk between leucocytes and the epithelia [41,42]. IL-22, a sibling of IL-17 [43], is mainly secreted by Th17 cells (a helper T lymphocyte subset) and an adaptive cytokine with reported dual-natured pro- and anti-inflammatory roles in the restitution of normal tissue and physiology after either sterile or non-sterile inflammation [44]. IL-22BP potently inhibits IL-22 biological effects (with much higher affinity than the IL-22 receptor) and is constitutively expressed in secondary lymphoid organs, breast, and epithelial tissues, preventing exaggerated effects (pathological inflammation) of IL-22 [40]. Also involved in TB and liver disease, IL-22 inhibits intracellular growth of *Mycobacterium tuberculosis* in human monocyte-derived macrophages [45] and is associated with acetaminophen-related hepatotoxicity and resolution of acute-on-chronic liver failure (ACLF) in small animal models [46,47]. Instead, IL-22 BP emerged as a novel marker in our score set. Consistently, in humans, IL-22BP was reported to be protective and in association with the development of and mortality from ACLF [48]. Nonetheless, the protective effect of IL-22BP should be further tested in large clinical studies.

At presentation with DILI, 84% of the TB patients had isolated elevation of liver enzymes, yet a high rate of survival. Elevation of liver enzymes in conjunction with jaundice (which was not seen in our study) is well established to reflect more severe DILI and a higher risk of adverse outcome [1] than isolated elevation of liver enzymes. Steuerwald et al. analysed serum taken within two weeks of clinical onset in 78 patients with acute DILI, without any description of causative drugs, and proposed that low values of four immune analytes (IL-9, IL-17, PDGF-bb, and RANTES) are predictive of early death [10]. In contrast, our study surveyed pre-treatment plasma samples and, therefore, cytokine profiles derived from our study may not be comparable to their findings. Our study results may be considered as a safety biomarker [49] to predict DILI, and whether these findings apply to other causative agents or diseases warrant further investigation.

Study limitations included no discrimination between idiosyncratic and dose-related hepatotoxicity; the nature of the post-Hoc analysis of prospective studies, in which patients were homogeneously selected by excluding clinically difficult cases, which may potentially limit the severity of DILI and render cytokine statistical analysis toward the null; data regarding the presence of slow acetylator phenotype/genotype of NAT2 gene or cytochrome P450 2E1 (CYP2E1) were not available.

In conclusion, we demonstrated a “safety” scoring of pre-treatment plasma biomarkers (IP-10, IL-22BP, and sCD163) predicting early DILI in patients with culture-confirmed pulmonary TB scheduled for TB medications. In contrast to IP-10 and sCD163, increased levels of IL-22BP may be associated with a protective effect against DILI development. For those with a higher risk of DILI, liver function tests should be performed more frequently for early diagnosis and management of DILI.

## Figures and Tables

**Figure 1 biomedicines-09-00891-f001:**
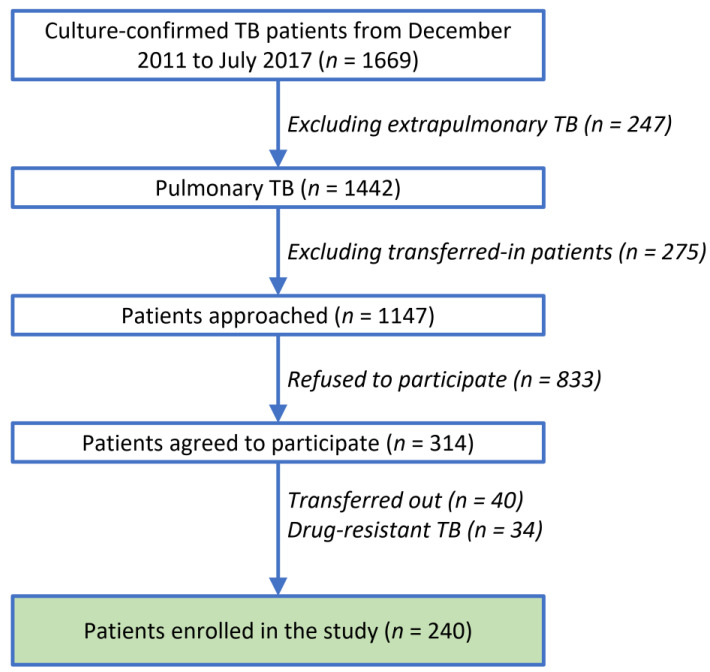
Patient flowchart.

**Figure 2 biomedicines-09-00891-f002:**
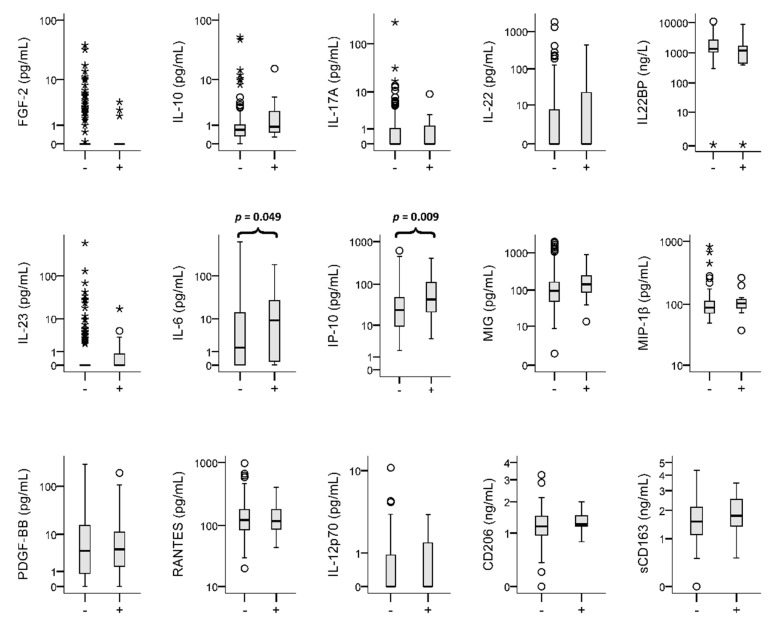
Box plots for cytokines. The box plot visualizes the following summary statistics: the middle line represents the median; the lower hinge corresponds to the first quartile (25th); the upper hinge corresponds to the third quartile (75th); upper and lower whiskers extend from the hinge respectively to the largest value and smallest value or no further than 1.5× interquartile range; data beyond the whiskers are outliers (circles) and extremes (asterisks). Outliers are at least 1.5 box lengths from the median and extremes are at least three box lengths from the median.

**Figure 3 biomedicines-09-00891-f003:**
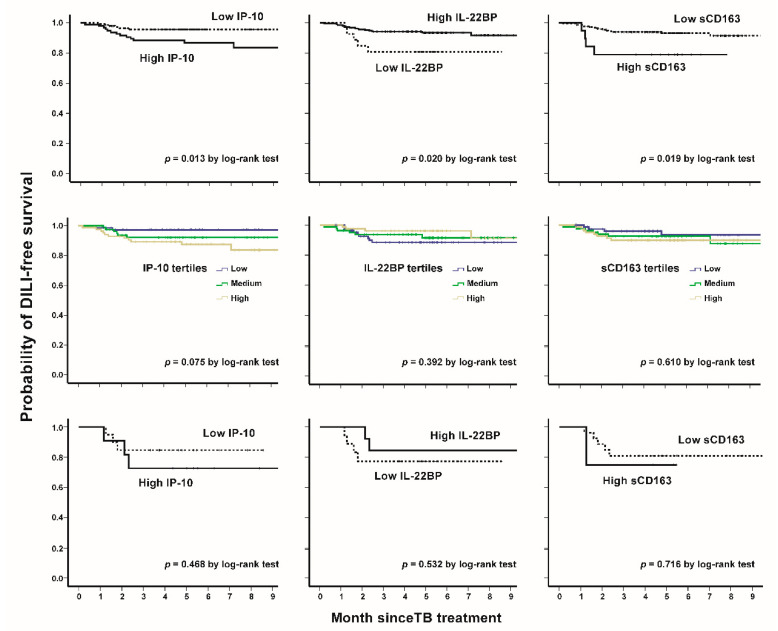
DILI-free survival curves, stratified by IP-10, IL-22BP, and sCD163 levels (**upper panel**); in tertiles (**middle panel**), and in the HBV subgroup (**lower panel**), respectively.

**Figure 4 biomedicines-09-00891-f004:**
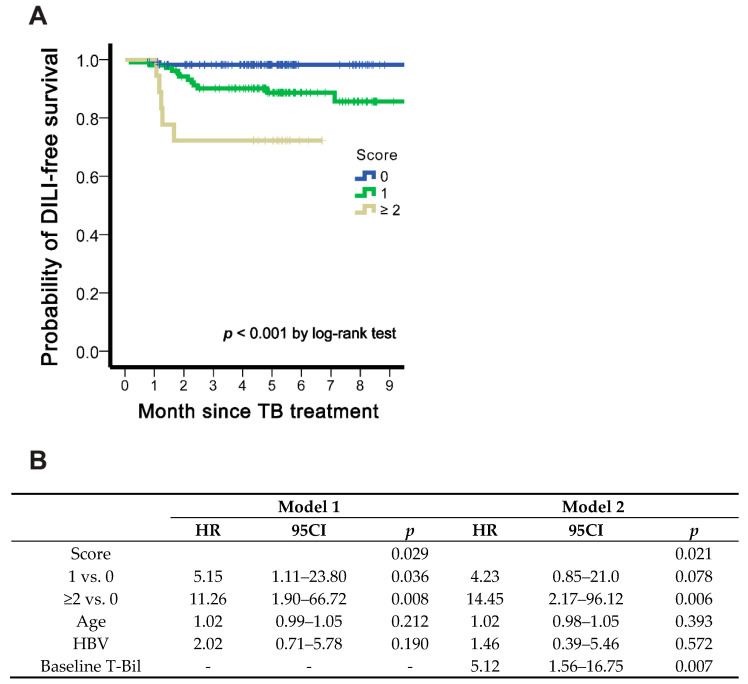
DILI-free survival in score stratification. (**A**) Kaplan Meier plot and (**B**) adjusted hazard ratios in cox regression analysis.

**Table 1 biomedicines-09-00891-t001:** Characteristics of patient demographics.

	All (*n* = 240)	DILI (*n* = 19)	Non-DILI (*n* = 221)	*p*
Age (years)	55.8 ± 17.7	63.9 ± 14.4	55.1 ± 17.8	0.021
Male sex (%)	143 (59.6)	12 (63.2)	131 (59.3)	0.812
BMI (kg/m^2^)	21.8 ± 3.3	21.7 ± 2.3	21.8 ± 3.4	0.798
Active smoking	39 (16.3)	6 (31.6)	33 (14.9)	0.099
Diabetes mellitus	38 (15.8)	3 (15.8)	35 (15.8)	1.000
Hypertension	34 (14.2)	2 (10.5)	32 (14.5)	1.000
Presence of cancer history	42 (17.5)	4 (21.1)	38 (17.2)	0.752
Hepatitis B virus	31 (12.9)	6 (31.6)	25 (11.3)	0.023
Hepatitis C virus	4 (1.7)	1 (5.3)	3 (1.4)	0.283
Haemodialysis	3 (1.3)	1 (5.3)	2 (0.9)	0.220
Autoimmune disease	10 (4.2)	1 (5.3)	9 (4.1)	0.569
Baseline AST (U/L)	26.5 ± 25.1	41.8 ± 46.7	25.1 ± 22.1	0.178
Baseline ALT (U/L)	22.4 ± 26.4	40.0 ± 52.0	20.9 ± 22.7	0.141
Baseline T-Bil (mg/dL)	0.6 ± 0.4	1.2 ± 1.1	0.6 ± 0.3	0.047
Abnormal baseline LFT	7 (2.9)	2 (10.5)	5 (2.3)	0.098

Abbreviations: ALT, alanine transaminase; AST, aspartic transaminase; BMI, body mass index; CI, confidence interval; DILI, drug-induced liver injury; HR, hazards ratio; LFT, liver function test; T-Bil, total bilirubin; Data were either mean ± standard deviation or number (%).

**Table 2 biomedicines-09-00891-t002:** Initial and peak liver profile as well as the Roussel Uclaf Causality Assessment Method (RUCAM) score of the 19 cases with drug-induced liver injury (DILI).

Case	Initial Data		Time to Peak (Days)	Peak Data		DILI Type	RUCAM Score	Probability of DILI
	ALT ≥ 2 ULN	Bil-T ≥ 2 mg/dL		ALT (ULN)	Bil-T (mg/dL)			
92.6/F	0	0	33	8.8	1.2	hepatocellular	6	probable
47.8/M	0	0	32	18.3	1.0	hepatocellular	7	probable
65.1/M	0	0	37	20.8	0.9	hepatocellular	8	probable
48.7/F *	0	0	38	10.9	0.5	hepatocellular	6	probable
72.4/F	0	0	50	8.6	0.6	hepatocellular	7	probable
60.4/F	0	0	24	5.6	0.3	hepatocellular	8	probable
65.5/F *	0	0	54	15.1	1.3	hepatocellular	8	probable
47.7/F	0	0	42	13.7	2.1	hepatocellular	8	probable
52.9/M *	0	0	35	7.0	0.9	hepatocellular	8	probable
66.3/M *	0	1	64	3.6	4.4	cholestatic	7	probable
60.1/M	1	0	55	7.3	1.3	hepatocellular	10	highly probable
64.1/M	0	0	55	5.4	0.5	hepatocellular	6	probable
91.0/M	0	0	25	1.8	5.0	mixed	8	probable
38.8/M *	0	0	48	6.3	0.7	hepatocellular	6	probable
85.5/M	1	1	6	55.8	4.1	hepatocellular	9	highly probable
58.5/M *	0	0	70	21.8	0.7	hepatocellular	6	probable
59.5/M	0	0	68	8.0	0.8	hepatocellular	6	probable
70.0/F	0	0	74	9.8	0.7	hepatocellular	12	highly probable
66.6/M	0	0	35	25.1	0.8	hepatocellular	8	probable

Abbreviations: ALT, alanine transaminase; Bil-T, total bilirubin; ULN, upper limit of normal; * hepatitis B virus carrier.

**Table 3 biomedicines-09-00891-t003:** Univariable and multivariable analyses of categorical cytokine risk factors for predicting DILI development in TB patients.

	Univariable			Multivariable Backward Selection		
	HR	95CI	*p*	HR	95% CI	*p*
IL-10	6.32	(0.84–47.5)	0.073			
IL-17A	1.22	(0.47–3.22)	0.683			
IL-22	1.68	(0.67–4.17)	0.267			
IL-22BP	0.32	(0.11–0.88)	0.027	0.20	(0.07–0.58)	0.003
IL-23	1.40	(0.51–3.90)	0.515			
IL-6	2.49	(1.00–6.20)	0.049			
IP-10	3.21	(1.22–8.44)	0.018	3.71	(1.35–10.21)	0.011
MIG	1.82	(0.53–6.25)	0.342			
MIP-1b	5.43	(0.73–40.7)	0.100	6.81	(0.87–53.47)	0.068
PDGF-BB	1.62	(0.47–5.57)	0.443			
RANTES	0.61	(0.08–4.54)	0.626			
IL-12p70	2.12	(0.83–5.38)	0.115			
CD206	4.02	(1.17–13.8)	0.027			
sCD163	3.47	(1.15–10.5)	0.028	3.28	(1.07–10.06)	0.038

Abbreviations: CI, confidence interval; DILI, drug-induced liver injury; HR, hazards ratio; IL, interleukin; IL-22BP, IL-22 binding protein; IP-10, interferon gamma-induced protein 10; MIG, monokine induced by interferon-gamma; MIP-1b, macrophage inflammatory protein-1beta; PDGF-BB, platelet-derived growth factor-BB; RANTES, Regulated upon Activation, Normal T Cell Expressed and Presumably Secreted; sCD163, soluble CD163; TB, tuberculosis.

**Table 4 biomedicines-09-00891-t004:** Patient characteristics, stratified by IP-10, IL-22 BP, and sCD163 levels.

	High IP-10 (*n* = 96)	Low IP-10 (*n* = 144)	*p*	High IL-22 BP (*n* = 213)	Low IL-22 BP (*n* = 27)	*p*	High sCD163 (*n* = 19)	Low sCD163 (*n* = 221)	*p*
Average age (years, SD)	60.8 ± 17.7	52.5 ± 16.9	<0.001	55.8 ± 18.2	56.2 ± 13.8	0.912	67.8 ± 12.6	54.8 ± 17.7	0.002
Male sex	61 (63.5)	82 (56.9)	0.348	122 (57.3)	21 (77.8)	0.059	11 (57.9)	132 (59.7)	>0.999
BMI, kg/m^2^ (mean, SD)	21.3 ± 3.3	22.1 ± 3.4	0.059	21.8 ± 3.4	22.2 ± 2.5	0.499	22.8 ± 2.9	21.7 ± 3.4	0.194
Active smoking	19 (20.4)	20 (14.0)	0.212	33 (15.8)	6 (22.2)	0.411	1 (5.6)	38 (17.4)	0.321
Diabetes mellitus	20 (20.8)	18 (12.5)	0.104	34 (16.0)	4 (14.8)	>0.999	4 (21.1)	34 (15.4)	0.514
Hypertension	21 (21.9)	13 (9.0)	0.008	34 (16.0)	0	0.018	5 (26.3)	29 (13.1)	0.160
Cancer status	19 (19.8)	23 (16.0)	0.490	38 (17.8)	4 (14.8)	>0.999	3 (15.8)	39 (17.6)	>0.999
Hepatitis B virus	11 (11.5)	20 (13.9)	0.696	13 (6.1)	18 (66.7)	<0.001	4 (21.1)	27 (12.2)	0.282
Hepatitis C virus	4 (4.2)	0	0.025	4 (1.9)	0	>0.999	0	4 (1.8)	>0.999
Haemodialysis	3 (3.1)	0	0.063	2 (0.9)	1 (3.7)	0.302	0	3 (1.4)	>0.999
Autoimmune disease	7 (7.3)	3 (2.1)	0.094	9 (4.2)	1 (3.7)	>0.999	0	10 (4.5)	>0.999
Baseline AST (U/L)	34.2 ± 37.6	21.9 ± 11.2	0.008	25.7 ± 25.5	32.1 ± 22.2	0.240	26.9 ± 14.6	26.4 ± 25.9	0.945
Baseline ALT (U/L)	27.4 ± 34.6	19.2 ± 18.8	0.039	21.4 ± 24.8	30.7 ± 36.3	0.092	24.9 ± 20.3	22.2 ± 26.9	0.664
Baseline T-Bil (mg/dL)	0.7 ± 0.6	0.6 ± 0.2	0.032	0.6 ± 0.5	0.7 ± 0.3	0.506	0.6 ± 0.3	0.6 ± 0.4	0.648
Abnormal baseline LFT	5 (5.2)	2 (1.4)	0.119	5 (2.3)	2 (7.4)	0.180	1 (5.3)	6 (2.7)	0.443
Drug-induced liver injury	13 (13.5)	6 (4.2)	0.013	14 (6.6)	5 (18.5)	0.047	4 (21.1)	15 (6.8)	0.051

Abbreviations: ALT, alanine transaminase; AST, aspartic transaminase; BMI, body mass index; IL-22BP, IL-22 binding protein; IP-10, interferon gamma-induced protein 10; LFT, liver function test; sCD163, soluble CD163; T-Bil, total bilirubin; Data were either mean ± standard deviation or number (%).

## Data Availability

The datasets used and analysed during the current study are available from the corresponding author upon reasonable request.

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
