# Peer review of "Circulatory Inflammatory Mediators in the Prediction of Anti-Tuberculous Drug-Induced Liver Injury Using RUCAM for Causality Assessment"

_biomedicines, 2021, doi:10.3390/biomedicines9080891_

Round 1
Reviewer 1 Report
Perfect revision. Check abbreviations, correct for:
RUCAM Roussel Uclaf Causality Assessment Method,
Reviewer 2 Report
The authors have made substantial revision to the presentation of the results, in light of the reviewers' comments. I do not have any further concerns.
This manuscript is a resubmission of an earlier submission. The following is a list of the peer review reports and author responses from that submission.
Round 1
Reviewer 1 Report
DILI by TB treatment is still a major clinical issue, as correctly mentioned by the authors. However, there is a significant problem of methodology.
Major points:
- It seems that data are left overs from a previous study, as mentioned in the text. Therefore, the submitted paper is not an original paper.
- This previous paper was likely not quoted in the reference list.
- Most disturbing is the lack of a robust causality assessment method like the RUCAM (updated version and published in 2016). This must be done, mentioned also in the abstract and title and under keywords. Details of the RUCAM evaluation must be presented. Make your paper more appealing to potential readers!
- Discuss and reference also that many DILI cases are not DILI but something else.
- Discuss and reference that RUCAM was appreciated in more than 80,000DILI cases published worldwide.
- Without RUCAM presented conconlusions are not acceptable.
- You should also reference and discuss the excellent DILI paper be Rathi, which uses RUCAM prospectively in his paper with TB patients, published in Annals of Hepatology some years ago.
Reviewer 2 Report
This prospective cohort study explores the association between pre-treatment serum markers and the likelihood of drug-induced liver injury with anti-tuberculosis regimen. From a cohort of 240 patients, 8% developed DILI, with advanced age and HBV carrier as pre-disposing factors. Positive correlation with DILI was established with high CD163, high IP-10 and low IL-22BP, presenting these as potential predictive markers for subsequent confirmation in other cohorts. Specific comments are as follow:
- Table 1, please clarify how the final parameter of abnormal baseline LFT is calculated? What do the numbers represent? This is not quite apparent from the methods.
- Given that baseline total bilirubin is also significantly different between DILI and non-DILI patients, was this adjusted for when determining the hazard ratio? It was mentioned in Line 212 that age and HBV status have been adjusted for, but there was no mention of baseline total bilirubin.
- Please clarify how the IP-10, IL-22BP and CD163 were dichotomized into high and low subgroups? What is the cut-off value for each of this case and what is rationale of setting this value?